# Nurses' Role in the Control and Treatment of Asthma in Adults: A Systematic Literature Review

Pedro Alexandre-Sousa [1,2,*] , Nuno Sousa [3] , Joana Bento [2] , Filipa Azevedo [2] , Maíra Assis [2] and José Mendes [4,5]

1   Center for Innovative Care and Health Technology (ciTechCare), 2400 Leiria, Portugal
2   Unidade Local de Saúde da Região de Leiria, 2400 Leiria, Portugal; jrbento@arscentro.min-saude.pt (J.B.); fsazevedo@arscentro.min-saude.pt (F.A.); mtassis@arscentro.min-saude.pt (M.A.)
3   Unidade Local de Saúde da Guarda, 6300 Guarda, Portugal; nuno.sousa@ulsguarda.min-saude.pt
4   INTELECTO—Psychology & Research, 9500 Ponta Delgada, Portugal; josemendes@intelecto.pt
5   Insight: Piaget Research Center for Ecological Human Development, 2805 Almada, Portugal
*   Correspondence: pesousa@arscentro.min-saude.pt

**Highlights:**

**What are the main findings?**

- Nurses play a crucial role in the control and treatment of asthma;
- Identify nurses' interventions for an asthma surveillance consultation in primary health care units.

**What is the implication of the main finding?**

- The integration of an asthma nursing consultation in PHC;
- Define the role of nurses as integrated providers of asthma-related care.

**Abstract:** Bronchial asthma is a chronic pathology and a global public health problem. However, asthma can be controlled and treated for the most part by patients, so the Portuguese General Directorate of Health recommends shared medical appointments in primary health care (PHC). The present study aims to identify the role of PHC nurses in the control and treatment of asthma in adults. Using the MeSH platform, the following descriptors were validated: asthma, nurses, adults. An individual search was carried out in the following databases: CINAHL (ESBSCO host), MEDLINE (Pubmed host), Web of Science, and Scopus. Out of a total of 280 publications, 79 of which were duplicates and 185 publications which did not meet the inclusion criteria, 16 publications remained readable. Of the eligible articles, there were 13 specialist reports, one mixed study, one quasi-experimental study, and one randomized trial. Education was the intervention most identified in the scientific evidence analyzed, and patient assessment, application of an asthma control questionnaire, verification and training of inhalation technique, empowerment for self-management of the disease, support, promotion of seasonal influenza vaccination, and use of written action plans were also identified. The results reveal that, although the scientific evidence on the intervention of these professionals is poorly developed, nurses play a crucial role in the control and treatment of asthma. The scientific evidence analyzed allowed the identification of interventions that can help the organization of a nursing health appointment, providing nurses with a crucial role in the control and treatment of asthma in adults in the context of PHC.

**Keywords:** asthma; nurses' role; adults; primary health care

## 1. Introduction

Bronchial asthma is a chronic pathology that affects all ages, with a worldwide prevalence of 9.1% in children, 11% in adolescents, and 6.6% in adults [1]. Hence, asthma is an important individual and public health problem at a global level, with an estimated 262 million people affected by this disease [2], having a significant impact on patients'

quality of life and healthcare systems [3–5]. The prevalence of asthma in Portugal is being studied, with the latest results pointing to a prevalence of 6.8% in 2010 [4].

Globally, it is estimated that 22.5 million disease-adjusted life years (DALYs) are lost annually due to asthma [6]. In Portugal, the number of years of life lived with disability (YLD) due to respiratory diseases is about 5%, with asthma being the main condition responsible for this (3.2%) [4].

Asthma is a respiratory disease usually associated with an organic hyper-response to direct or indirect stimuli, presenting as chronic inflammation of the airways, persisting even when lung function is normal and there are no symptoms, but which can be normalized with medication [5]. It is a pathology characterized by symptoms such as wheezing, paroxysmal dyspnea, a feeling of tightness in the chest, cough, or increased expiratory time, and the intensity of these symptoms may vary over time [3–5]. These variations are often triggered by factors such as physical exercise, exposure to irritants or allergens, temperature changes, or respiratory infections [5,7]. Asthma can be controlled and treated in most patients, allowing (i) a reduction in the need for reliever medication, (ii) patients that can lead productive and active lives, (iii) the maintenance of near-normal lung function, and (iv) prevention of exacerbations [5]. In this sense, there is evidence that asthma self-management can reduce the use of health services and school/job absenteeism, thus justifying the need to create specific asthma programs [2,8]. Therefore, in 2018, the Portuguese Directorate-General for Health (DGH) issued the Integrated Care Plan (ICP) for Asthma in Children and Adults (aged 14 years and over), which aims to guide and organize clinical practice to improve results and quality. This ICP also aims to achieve the most effective use of resources, emphasizing activities that are thought to affect the quality of care for this pathology [4].

Despite advances in the knowledge of its pathophysiological mechanisms and the existence of several effective therapeutic options for the control of asthma, the goals of treatment are not always achieved [3,5]. Given the importance of asthma monitoring and control, the DGH recommends primary health care (PHC) consultations, where family health units (FHUs) are in a privileged context to structure a consultation based on the ICP and scientific evidence [4]. In this sense, the present systematic review of the literature (SRL) aims to identify the role of PHC nurses in the control and treatment of asthma in adults.

## 2. Materials and Methods

The SLR protocol was registered with PROSPERO with the code CRD42023455434, and was guided according to the methodological guidelines proposed by the Joanne Briggs Institute (JBI) [9]. As shown in Table 1, the mnemonic PIO (population, interest, and outcome) was considered when constructing the investigation question [10]: What is the role (I) of the PHC nurse (P) in the control and treatment of asthma in adults (O)?

**Table 1.** Mnemonic used in this study.

| Criteria | Description | Current Study Implementation |
|---|---|---|
| P | Population | PHC nurses who perform asthma consultations |
| I | Interest | PHC nurses' role in asthma consultations |
| O | Outcome | Outcome of nurses' interventions in the control and treatment of asthma |

The literature search was carried out based on the descriptors validated in the Medical subject heading platform (MeSH). Considering CINAHL (ESBSCO host), MEDLINE (Pubmed host), Web of Science, and Scopus, we searched for full-text articles in Portuguese, Spanish, and English. To create the search expression, the Boolean operators "AND" and "NOT" were used: asthma AND "nurse* role" NOT child*.

The present study did not consider a specific time frame, as the historical milestones of nurses' involvement in the consultation of asthma patients remain unknown. The inclusion and exclusion criteria adopted are presented in Table 2. Duplicate publications, and studies that did not address the theme relevant to the objective of this study and that used methods other than those selected were excluded.

**Table 2.** Inclusion and exclusion criteria.

| | Criteria | Description |
|---|---|---|
| Inclusion Criteria | C1 | Primary studies, reviews, and specialist reports |
| | C2 | The article must contain nursing interventions for asthma treatment |
| | C3 | Nursing interventions may be used in PHC context |
| | C4 | Nursing interventions are directed to adults |
| Exclusion Criteria | C1 | Articles witch not in English, Portuguese, or Spanish |
| | C2 | The article does not present the full text |
| | C3 | Does not present nursing interventions for asthma treatment |
| | C4 | Nursing interventions found cannot be used in PHC context |
| | C5 | Nursing interventions can only be used on children |
| | C6 | The article does not meet the minimum methodological quality assessment assessed by the JBI Appraisal Tools |

The searches were conducted on 18 November 2023. Both the research and the review of the articles were carried out by at least two independent reviewers.

Through the search of scientific databases, 280 articles were obtained, of which 71 duplicates were excluded. The process of identification and selection of articles took place in two stages. The first stage consisted of screening the titles and abstracts according to the inclusion and exclusion criteria. Upon this screening, 162 were rejected due to their failure to meet the previously established inclusion criteria. Subsequently, the articles resulting from this first analysis were fully evaluated. The full and critical reading of the articles was evaluated, taking into account the process of evaluation and interpretation of the evidence, as well as its validity and relevance, following the methodology proposed by the JBI appraisal tools [11]. These tools were applied by two independent reviewers, and articles were included, or excluded, if both had the same opinion. If there was a difference of opinion between both reviewers regarding the quality of the article, a third reviewer would be asked to assess the quality using the same methodology. Figure 1 presents a diagram according to Preferred Reporting Items for Systematic Reviews and Meta-Analysis (PRISMA) [12], which shows the total number of references returned by the search and those excluded at each step. After the selection and evaluation process, 16 registries were integrated into the present study.

The articles included in the review were summarized and presented in a table prepared by the authors.

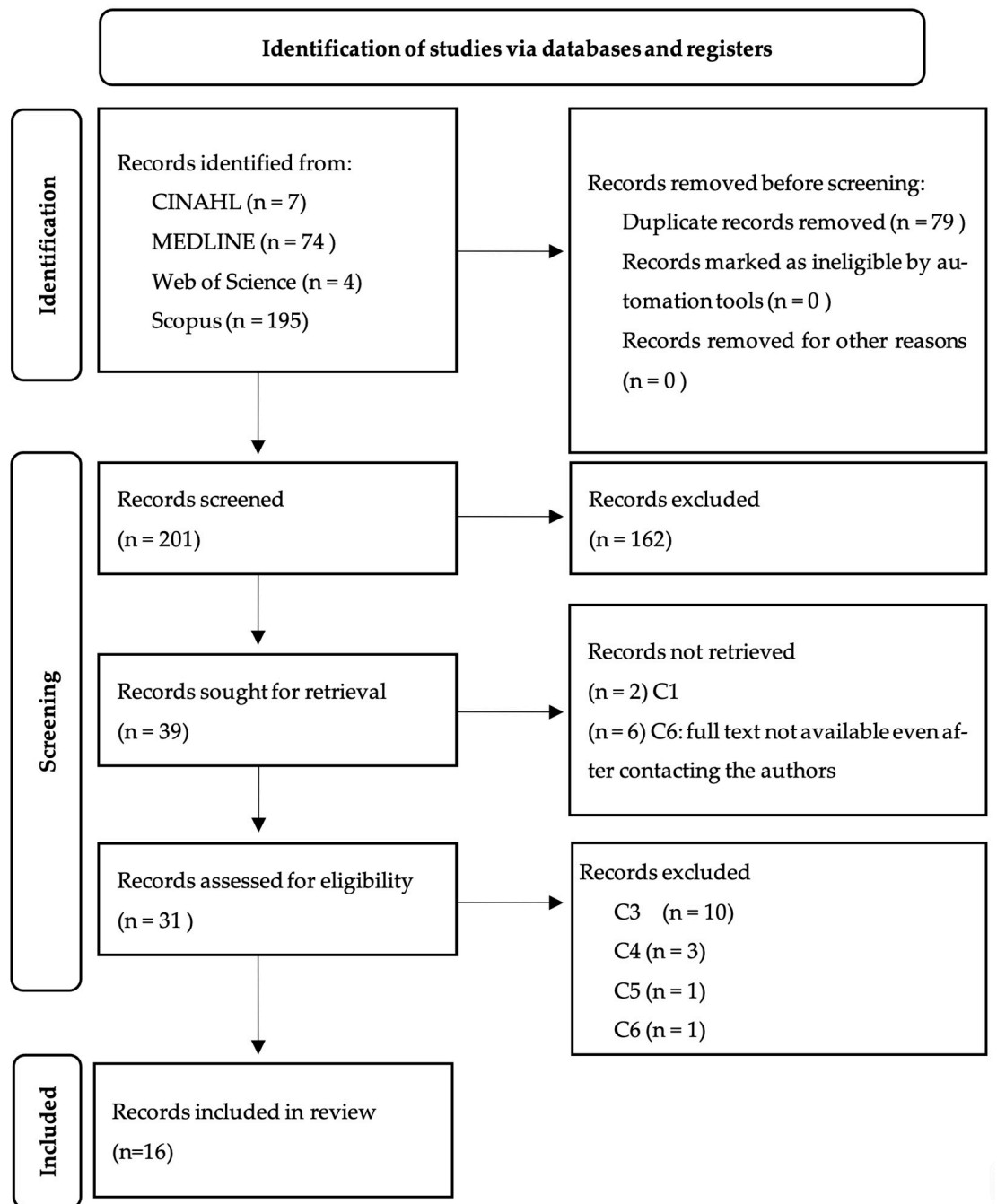

**Figure 1.** Preferred Reporting Items for Systematic Reviews and Meta-Analysis diagram.

### 3. Results

Of the selected articles, 13 specialist reports were identified, of which eight were published in the United Kingdom (UK) and five in the United States of America (USA). We also identified a mixed descriptive study conducted in the United Kingdom, a quasi-experimental study developed in Jordan, and finally, a randomized experimental study developed in Turkey. Table 3 shows the objectives and participants involved in each study, the level of evidence, the nursing interventions that can be performed in the context of asthma consultations in PHC, and the results in regard to the control and treatment of this pathology.

**Table 3.** Presentation of the studies included in the review.

| Authors and Year | Title | Type and Country | Objectives and Participants | Evidence Level | Nursing Interventions | Outcomes |
|---|---|---|---|---|---|---|
| Robertson et al. (1997) [13] | Adult asthma review in general practice: Nurses' perception of their role | Mixed descriptive study<br><br>UK | To assess the level of asthma training of nurses who care for adults with asthma and to verify whether this interferes with the perception of their role.<br><br>167 nurses of the Grampian Nurse Association | IV | Patient education;<br>Discussion and support of the patient's concerns and anxieties;<br>Guidance of the patient for guided self-management;<br>To evaluate asthma control according to the following parameters: (i) nocturnal awakenings, (ii) limitation of asthma-related activity, (iii) frequency of use of short-acting bronchodilators;<br>Assess peak expiratory flow (PEF);<br>Review of the inhalation technique with the patient;<br>Case management. | Reduces the morbidity of asthma patients and improves their quality of life. |
| Barnes (1998) [14] | The nurse's role in asthma management | Specialist report<br><br>UK | To summarize the current roles of nurses in asthma care and highlight areas that need to be addressed in the future. | V | Ask patients about their medical history;<br>Check the inhalation technique;<br>Evaluate PEF;<br>Educating the patient about asthma;<br>Provide explanatory literature;<br>Formulate a structured treatment plan together with the doctor and the patient;<br>Carry out regular consultations;<br>Advise the patient over the phone, if necessary. | Improves the results of asthma treatment |
| Coakley (2001) [15] | Helping patients to master correct inhaler techniques: nursing role | Specialist report<br><br>UK | To provide nurses not only with inhalation techniques, but also how to identify areas of difficulty and find appropriate solutions. | V | Take into account the specificities of elderly patients when using inhalers;<br>Demonstrate and train in the use of inhalers;<br>Assess the need for the use of an expander chamber. | Prevents errors in inhalation technique; Expander chambers make the inhaler more accessible for the elderly. |

**Table 3.** *Cont.*

| Authors and Year | Title | Type and Country | Objectives and Participants | Evidence Level | Nursing Interventions | Outcomes |
|---|---|---|---|---|---|---|
| Jordan and White (2001) [16] | Bronchodilators: implications for nursing practice | Specialist report UK | To update knowledge and encourage the review of practice in the area of bronchodilators. | V | Regularly monitor asthma patients; Educate the patient about bronchodilators: (i) mechanism of action, (ii) adverse effects, and (iii) correct use; Recommend the use of an expanding chamber when the inhalation technique is not effective; Assess for side effects to bronchodilators; Recommend mouthwash after using bronchodilators; Recommend breathing exercises; Ensure that the patient receives a written action plan. | Improves patient care |
| Brooks and Hayden (2003) [17] | Exercise-induced asthma | Specialist report USA | To present the pathophysiology of exercise-induced asthma, the treatment and the role of the nurse in its management. | V | Assess vital signs and baseline spirometry; Educate the patient/family about: (i) avoidance of high-risk environments, (ii) inadequate heating/cooling when exercising, (iii) pre-treatment or symptomatic treatment with prescribed medications, and (iv) correct use and inhalation technique of medications; To evaluate the implications of physical exercise limitations on the patient's expectations and quality of life. | Prevents serious sequelae and allows the asthma patient to participate in any sporting level. |
| Huss et al. (2003) [18] | General principles of asthma management: environmental control | Specialist report USA | To educate nurses on the control of environmental factors that contribute to asthma triggers. | V | Educate the patient/family on: (i) avoiding tobacco smoke, (ii) control of dust mites and house dust and (iii) control of other potential allergens (e.g., cat, dog, cockroaches, fungal spores); Assist patients with essential environmental control measures. | Symptom relief; Reduction of environmental stimuli that trigger asthma attacks. |

**Table 3.** *Cont.*

| Authors and Year | Title | Type and Country | Objectives and Participants | Evidence Level | Nursing Interventions | Outcomes |
|---|---|---|---|---|---|---|
| Kallenbach et al. (2003) [19] | Process Improvement for Asthma: An Integrated Approach | Specialist report USA | To describe a program designed to improve the care delivery for people with asthma | V | Promote the use of asthma plans with: (i) daily management, (ii) medications, (iii) how to deal with an acute asthma episode, (iv) checklists (e.g., lists of exacerbation triggers and early asthma symptoms), (v) personal record of better PEF, (iv) medication regimen, (vii) patient action in case of exacerbation; Promote influenza vaccination; Patient education; Promote the role of asthma coordinator in the team. | Improves the delivery of asthma care |
| Ramirez (2003) [20] | Management of asthma emergencies | Specialist report USA | To describe the role of nurses in providing emergency care to people with asthma throughout the lifespan. | V | Evaluate all episodes of asthma exacerbation to determine their severity; Obtain a history on: (i) triggers, (ii) past exacerbation experiences and (iii) drug effects; Assess vital signs; Assess PEF or forced expiratory volume in 1 s (FEV1); Require medical collaboration; Administer the therapy according to the clinical prescription; Educate the patient about: (i) reliever medication, (ii) self-control of asthma at home if it is of low severity, (iii) prevention of stimuli that lead to exacerbation. | Preventing and controlling asthma exacerbation |

**Table 3.** *Cont.*

| Authors and Year | Title | Type and Country | Objectives and Participants | Evidence Level | Nursing Interventions | Outcomes |
|---|---|---|---|---|---|---|
| McAllister (2004) [21] | An overview of the current asthma disease management guidance | Specialist report<br><br>UK | To provide evidence-based guidelines for health care in relation to asthma. | V | Assess asthma control in the last week/month regarding: (i) nocturnal awakenings due to asthma symptoms, (ii) daytime symptoms, (iii) interference with daily activities and iv) exacerbations/use of reliever bronchodilator;<br>Teach patients to: (i) recognize their own triggers, (ii) avoid symptoms, (iii) self-manage asthma, (iv) sanitize the expanding chambers;<br>Verify the patient's inhalation technique;<br>Assess patient adherence;<br>Promote the use of written action plans;<br>Empower patients to take control of their disease. | Improves asthma control; Empowers the patient to cope with the responsibility for the day-to-day treatment of their asthma. |
| Newell (2006) [22] | Concordance with asthma medication: the nurse's role | Specialist report<br><br>UK | To examine the non-adherence to treatment by the patient with asthma and to discuss the role of the nurse in improving adherence to the medication regimen. | V | Empower patients for self-management;<br>Use asthma action plans with the following characteristics: (i) written, (ii) individualized, (iii) with the patient's agreement, (iv) with records of prescribed medications and how/when to take them;<br>Provide information on: (i) how to recognize when asthma is getting out of control, (ii) how and when to adjust medication, (iii) when to seek medical advice, (iv) how to manage severe attacks, (v) side effects of prescribed medication;<br>Suggest practical solutions that help the patient adhere to the medication regimen in their daily routine;<br>Encourage patients to rinse with water after inhaling corticosteroids to reduce mouth deposition;<br>Recommend the storage of powder devices in dry environments. | Improves patient self-management and adherence to the medication regimen; Prevents potential problems of dysphonia and oral candidiasis; Increases the effectiveness of the medication. |

**Table 3.** *Cont.*

| Authors and Year | Title | Type and Country | Objectives and Participants | Evidence Level | Nursing Interventions | Outcomes |
|---|---|---|---|---|---|---|
| Sims (2006) [23] | An Overview of Asthma | Specialist report USA | To provide an overview of asthma, its pathophysiology, signs and symptoms, and their management. | V | Monitor the patient's oxygen saturation and administer supplemental oxygen as directed; Position the patient in a high Fowler position and slightly leaning forward; Continuously monitor the patient's respiratory and cardiovascular status; Assess PEF frequently; Administer prescribed medication and observe its effects; Assist the patient in a calm and reassuring way; Educate the patient (and/or his/her family) on: (i) appropriate use of medications, (ii) appropriate use of a metered-dose inhaler/nebulizer, (iii) signs and symptoms of an exacerbation, (iv) monitoring, normal PEF values and measures to be taken if values decrease, (v) understanding of the treatment plan, (vi) increased fluid intake to decrease secretions, (vii) adequate technique for contracted lip breathing, (viii) adequate technique for diaphragmatic breathing, (ix) identification of factors that trigger crises, (x) deep breathing exercises to prevent atelectasis, xi) environmental control, (xii) good health practices, such as exercise, rest and nutrition, (xiii) relaxation techniques. | Monitors the severity of the episode and response to treatment; Prevents cardiac arrhythmias that can lead to cardiovascular collapse and death; Prevents respiratory failure; Reduces exacerbations and hospitalizations. |
| Booker (2007) [24] | Peak expiratory flow measurement | Specialist report UK | To describe how nurses can help asthma patients obtain accurate PEF measurements. | V | Assess PEF in asthma visits; Instruct and demonstrate to patients on the use of PEF devices; When equipment is used between patients, comply with cleaning and disinfection procedures and ensure the use of disposable unidirectional nozzles; Promote the replacement of meters annually. | PEF measurement is useful in diagnosing and monitoring patients with asthma. |

**Table 3.** *Cont.*

| Authors and Year | Title | Type and Country | Objectives and Participants | Evidence Level | Nursing Interventions | Outcomes |
|---|---|---|---|---|---|---|
| Mendes (2015) [25] | Providing meaningful support to patients with asthma | Specialist report<br><br>UK | To raise awareness of asthma and the role that nurses can play in the management of this disease. | V | Provide appropriate guidance on how to avoid environmental triggers and help patients self-manage;<br>Check the inhalation technique;<br>Provide a personal asthma action plan that includes: (i) the triggers of exacerbations, (ii) current treatment, (iii) how to prevent relapse and iv) when and how to seek emergency help;<br>Promote mental health. | Improves asthma control |
| Al-Kalaldeh et al. (2016) [26] | Effectiveness of Nurse-Driven Inhaler Education on Inhaler Proficiency and Compliance Among Obstructive Lung Disease Patients: A Quasi-Experimental Study | Quasi-experimental<br><br>Jordan | To assess the impact of inhaler education by nurses on inhaler adherence and proficiency.<br><br>Participants: 121 people aged 18 years or older, diagnosed with obstructive pulmonary disease (asthma and COPD) for more than two years and treated with inhaled therapy. | III | Verify the inhalation technique performed by the patients;<br>Educate and train inhalation technique. | Increases the ability to perform the inhalation technique;<br>Patients' adherence to prescribed doses, timing and frequency of inhaler use increases. |
| Scullion (2018) [27] | The Nurse Practitioners' Perspective on Inhaler Education in Asthma and Chronic Obstructive Pulmonary Disease | Specialist report<br><br>UK | To examine, from the perspective of nurses, the role they play in the management of asthma and the interventions they can undertake to improve outcomes for patients with asthma, with a particular focus on inhaled education. | V | Evaluate common patient errors in inhalation technique;<br>Educate the patient on the correct inhalation technique;<br>Communicate with patients to better understand their motivations, concerns, and preferences;<br>Advise and explain about the importance of adherence. | Improves adherence;<br>Prevents exacerbations;<br>Improves the self-management of the asthma patient. |

**Table 3.** *Cont.*

| Authors and Year | Title | Type and Country | Objectives and Participants | Evidence Level | Nursing Interventions | Outcomes |
|---|---|---|---|---|---|---|
| Şanlıtürk and Ayaz-Alkaya (2023) [28] | The effect of a nurse-led home visit program on the care burden of caregivers of adults with asthma: A randomized controlled trial | Randomized controlled trial<br><br>Turkey | To determine the effect of a home visiting program on the perceived burden of care by family caregivers of adults with asthma.<br><br>Participants: 130 family caregivers of adults with asthma enrolled in an outpatient pulmonology consultation at a hospital. | I | Educate about (i) general information about asthma, (ii) treatment methods, (iii) disease management, (iv) proper administration of medications, (v) asthma triggers, and (vi) ways to reduce exposures that cause exacerbations;<br>Demonstrate and train the use of inhalers;<br>Advise;<br>Conduct home visits;<br>Involve the family in care;<br>Support the family. | Improves self-efficacy and decreases the caregiving burden perceived by caregivers. |

## 4. Discussion

Through this RSL, it was possible to identify that the scientific evidence relating to the intervention of nurses in asthma consultations is poorly developed, with only three primary research studies identified and the remaining articles considered specialists' reports. It is possible to verify that most of the expert reports came from the UK. It is possible that these data are due to the higher prevalence of asthma in this country (17.6%) compared to other European countries [29].

The evidence gathered from the articles analyzed allowed us to identify that nurses play a crucial role in the control and treatment of asthma [13–28], as there is substantial evidence that supports systematic follow-up or structured consultations on asthma by nurses [13–15,19,25,28].

Of the interventions identified in this review, the largest consensus was on the education of individuals with asthma [13,14,16–28]. In this sense, the following topics that nurses should address in an asthma-related consultation stand out: (i) general information about asthma [14,28], (ii) medication regimen [16,17,19,20,22,23,27,28], (iii) environmental control [17,18,28], (iv) prevention and management of exacerbations [19,21,23,25,28], (v) self-management of the disease [19–22,25], (vi) when to seek out health services [22,25], (vii) understanding the treatment plan [19,23], (viii) breathing exercises [16,23], (ix) use of PEF devices [23,24] and (x) inhalation technique [13–15,17,21,23,25–28].

Given the importance of the correct use of medication for asthma management, the articles analyzed reinforce that the role of nurses should go beyond simple education, and also include the verification and training of inhalation technique [13–17,21–23,25–28]. Therefore, the evaluation of the need to use an expansion chamber in the administration of inhaled medication was also pointed out as a role of the nurse when verifying the inhalation technique [15,16,21]. It is important to emphasize that, for the satisfactory efficacy of medication, it is necessary not only to verify the inhalation technique but also to promote adherence to the therapeutic regimen [13,21,22,27].

Several studies analyzed show the importance of the role of nurses in empowering individuals with asthma to self-manage the disease [13,21–23,25,27], corroborating the B2 criterion of the Regulation of the Competence Profile of General Care Nurses [30] and the application of Orem's self-care theory in patients with asthma [31]. It was also found to be useful to involve the family in their care [17,18,23,28] and/or home visits [28]. For self-management of the disease by individuals with asthma, it is essential to use written action plans carried out jointly between the health team and the patient [14,16,19,21,22,25]. This evidence is in line with what is recommended by the DGH [4] and by the Global Initiative for Asthma (GINA) [5].

Both DGH [4] and GINA [5] advocate the use of an asthma control assessment scale, such as the Asthma Control Test (ACT) [32]. This fact is corroborated by this study [13,14,21]. The ACT has a higher sensitivity than the Asthma Control Questionnaire-7 (ACQ-7) when evaluating the asthma control criteria recommended by the GINA [33], making it a good option for application in asthma consultations [34].

The results also reveal that the peak expiratory flow (PEF) was pointed out by the analyzed articles as being important for the evaluation of people with asthma [13,14,17,20,23,24]. This is an intervention that can be performed by nurses, following the most recent recommendations of the GINA [34]. The assessment of vital signs is also the role of nurses and is useful in the assessment of people with asthma, especially body mass index (BMI), blood pressure (BP), heart rate (HR) and, in case of exacerbation, peripheral blood oxygen saturation ($SpO_2$) [17,20,23] interventions are also recommended by the DGH [4].

Another role evidenced by the articles analyzed was the establishment of a supportive therapeutic relationship between nurses and asthma patients [13,22,25,27,28]. In this sense, one SRL identified communication, trust, and follow-up over time as essential elements of support for people with asthma [35]. Regular support from health professionals for people with asthma reduces the need for them to use health services and improves their

quality of life [36], with communication and effective interpersonal relationships being key competencies of nurses [30].

Furthermore, only one of the articles analyzed referred to the importance of seasonal influenza vaccination [19], despite the recommendations of the DGH [4] and GINA [5] regarding this vaccine for people with asthma, as well as several other studies [37,38].

In addition to the surveillance consultation, nurses also play an important role in the management of asthma-related emergencies [20,23]. In this sense, the following interventions, which should be performed by nurses in consultations for acute illnesses related to asthma complications, were identified: (i) assessment of vital signs [17,20,23], (ii) evaluation of PEF [13,14,17,20,23,24], (iii) oxygen administration [23], (iv) administration of medication prescribed by the physician [20,23], and (v) patient education on crisis management and relapse prevention [19–23,25,28].

Finally, the studies analyzed also show that integrating the role of the nurse with the role of the physician in the management of asthma results in significant benefits for asthma patients [13,17,19,23,25], in addition to better cost-effectiveness for the health system [22,25], which is corroborated by an SRL carried out by Htay and Whitehead [39].

### 4.1. Limitations of the Study

One of the biggest risks when conducting an SRL is the definition of the research strategy [9,10]. In this sense, several free searches were carried out to improve the search terms and Boolean operators that were used, maintaining a balance between the sensitivity and relevance of the results. Different databases were used to minimize the risk of losing relevant information. Another limitation of the present study was the linguistic restriction to Portuguese, English, and Spanish, excluding studies published in other languages. In addition, the scientific evidence of most of the selected articles is limited, with only three studies and 13 expert reports found. Further studies evaluating the effectiveness of nursing interventions for controlling asthma are recommended. Of equal relevance would be assessing the knowledge and perception of Portuguese nurses about the performance of this role in asthma control.

### 4.2. Implications for Clinical Practice

As most of the studies analyzed support the role of nurses as integrated providers of asthma-related care, and this practice is not yet common in Portuguese PHC, this SRL has several implications for practice. The integration of an asthma nursing consultation in PHC (i) helps to alleviate the burden currently placed on physicians, (ii) increases accessibility for patients, (iii) has positive implications for the cost of health care, as it appears that multidisciplinary asthma management helps to reduce costs and improve the efficiency of the health care system, (iii) improves asthma control, and (iv) empowers patients to better manage their disease.

### 5. Conclusions

This study revealed that PHC nurses play a crucial role in the management and treatment of asthma in adults. Data analysis allowed us to find nursing interventions for asthma control, highlighted in several reports, standards, and studies. Thus, the contribution of this study makes it possible to organize a regular surveillance nursing consultation in PHC units. The role of nurses is also highlighted in (i) the assessment of the patient, including BMI, vital signs, and PEF; (ii) the assessment of asthma control by a questionnaire, such as the ACT; (iii) the verification and training of the inhalation technique, including assessment of the need for the use of expander chamber devices; (iv) in the education of the patient and their family, (v) in the evaluation and promotion of adherence to the management of the therapeutic regimen, (vi) empowering people with asthma to self-manage the disease, (vii) support for the person with asthma through a therapeutic relationship of trust and support in the event of any doubts or the need for care, (viii) promoting adherence to seasonal influenza vaccination, and (ix) the use of written

action plans to be drawn up jointly by nurses, doctors, and patients, and delivered to the patient at the end of consultations. On the other hand, the role of nurses in acute asthma-related illness consultations are described as (i) assessment of vital signs and PEF, (ii) administration of oxygen and other medication prescribed by the physician, and (iii) patient education on crisis management and relapse prevention. Multidisciplinary asthma management programs in PHC show both health gains for patients and cost-effectiveness for health services.

**Author Contributions:** Conceptualization, P.A.-S., J.B., N.S., F.A., M.A. and J.M.; methodology, P.A.-S. and J.M.; validation, P.A.-S., J.B., N.S., F.A., M.A. and J.M.; formal analysis, P.A.-S., J.B., N.S., F.A., M.A. and J.M.; writing—original draft preparation, P.A.-S. and J.M..; writing—review and editing, P.A.-S., J.B., N.S., F.A., M.A. and J.M.; visualization, P.A.-S., J.B., N.S., F.A., M.A. and J.M.; supervision, J.M; project administration, P.A.-S. All authors have read and agreed to the published version of the manuscript.

**Funding:** This research received no external funding.

**Institutional Review Board Statement:** Not applicable.

**Informed Consent Statement:** Not applicable.

**Data Availability Statement:** Data are contained within the article.

**Conflicts of Interest:** The authors declare no conflicts of interest.

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
