# Peer review of "Nurses’ Role in the Control and Treatment of Asthma in Adults: A Systematic Literature Review"

_arm, doi:10.3390/arm92030019_

Round 1
Reviewer 1 Report
Comments and Suggestions for Authors
This is a well-written paper, because the topic quite draws the readers’ attention, and asthma prevention is an important issue, no matter regarding public health or social science perspective. In general, this paper is organized, though its outcome is not specifically inspiring, due to the content collected with 13 special reports from the UK and the USA, not very evidence-based and many article sorting results are from quite some period ago, not able to represent the currently update medication and school health, it could be quite difficult to collect those valuable raw data. There are a few suggestions that need to be considered. For example, the authors may describe more in detail how the process of quality of the papers between the two reviewers' agreement in the methodology session, in addition, the PICO inclusion criteria could be broader to collect more good papers for further reviewing.
Author Response
Dear reviewer, thank you for your suggestions/concerns.
We add information about the agreement between the two reviewers to the paper (lines 115-118). All changes made to the paper regarding both reviewers' suggestions are highlighted in blue color.
Your comments are very relevant since, during the article's writing, we were aware of the concerns you raised with us. However, considering the objective of the study (the role of the nurse in an asthma assessment consultation in adults in the context of PHC), scientific evidence besides what was found is poorly known. It should be noted that the research strategy and the PICO mnemonic took into account the experience of several free searches. We used the search strategy considered to be the most balanced between the number of articles found and their relevance to answering our research question.
We hope we answered your concerns, and we remain available for further improvements if you think it's needed.

Reviewer 2 Report
Comments and Suggestions for Authors
1. Line 52-54: it is stated that YLD of respiratory diseases corresponds to a proportion of 5%. The question is, as YLD presents in number of years lived with disability, the presented proportion is the proportion of respiratory diseases to what?
Same for asthma.
2. I believe the Propspero registration code is better to be in the methods section.
3. Inclusion criteria 1: did you include reviews?
Additionally, what do you mean by specialist’s/specialty reports? Is it guidelines? Or reviews? As you want to define the role of nurses, I think neither guidelines nor reviews are not appropriate to include as they are summarizing previous data. I believe you should evaluate their references instead of including them in your study.
4. In figure 1, the number of reports assessed for eligibility is 31, the number of reports excluded is (10+3+1) 14, the number of studies included is 16. These numbers do not match.
5. In figure 1, instead of “reports” it’s better to use studies, or articles.
6. Please provide details on quality assessment.
Comments on the Quality of English Language1. There are some grammars and editing mistakes:
a. Line 1: correct “Sistematic Review” to “Systematic Review”
b. Line 10-21: The highlights section is the template! I believe authors forgot to edit or delete this part.
c. Line 33-34: “support” is repeated.
d. Different sentences are in Spanish/Portuguese; line 48-50, line 96-97,
e. Line 112: at the end of the line “because due”, both words have the same meaning. One should be deleted.
f. Expert reports, specialists’ reports, and specialty reports are used interchangeably. I suggest using one word for all of them.
g. The whole manuscript needs to be edited grammatically.
Author Response
Dear reviewer, thank you for your comments and recommendations. They were useful to improve the paper. The new version of the paper is attached and all alterations were highlighted in blue.
Addressing your comments:
- Line 52-54: it is stated that YLD of respiratory diseases corresponds to a proportion of 5%. The question is, as YLD presents in number of years lived with disability, the presented proportion is the proportion of respiratory diseases to what? Same for asthma.
5% of all YLD are related to respiratory diseases
3.2% of all YLD are related to asthma
- I believe the Prospero registration code is better to be in the methods section.
Done
- Inclusion criteria 1: did you include reviews?
Yes, as stated in inclusion criteria C1
Additionally, what do you mean by specialist’s/specialty reports? Is it guidelines? Or reviews? As you want to define the role of nurses, I think neither guidelines nor reviews are not appropriate to include as they are summarizing previous data. I believe you should evaluate their references instead of including them in your study.
Thank you for your comment. Although on the bottom of the evidence pyramid, we consider including specialist’s reports (also known as expert’s opinion) due to the lack of higher evidence studies to answer the research question. All of these reports were published in peer-reviewed journals. We acknowledged it as a limitation of our study.
- In figure 1, the number of reports assessed for eligibility is 31, the number of reports excluded is (10+3+1) 14, the number of studies included is 16. These numbers do not match.
Thank you for noticing it. The size of the last “excluded reports” box was insufficient to show all the text. We changed it, so 31-10-3-1-1=16.
In figure 1, instead of “reports” it’s better to use studies, or articles.
Replaced “reports” for “records” as recommended by PRISMA.
- Please provide details on quality assessment.+
Information added to the text (115-118).
Comments on the Quality of English Language
- There are some grammars and editing mistakes:
- Line 1: correct “Sistematic Review” to “Systematic Review”
Corrected
- Line 10-21: The highlights section is the template! I believe authors forgot to edit or delete this part.
Edited
- Line 33-34: “support” is repeated.
Corrected
- Different sentences are in Spanish/Portuguese; lines 48-50, lines 96-97,
Corrected
- Line 112: at the end of the line “because due”, both words have the same meaning. One should be deleted.
Corrected
- Expert reports, specialists’ reports, and specialty reports are used interchangeably. I suggest using one word for all of them.
Done
- The whole manuscript needs to be edited grammatically.
The whole manuscript was reviewed grammatically. Changes are shown in blue color.
We hope we answered your concerns, and we remain available for further improvements if you think it's needed.

Round 2
Reviewer 2 Report
Comments and Suggestions for Authors
Thank you for making the points clear